# Cohort profile: The Clinical and Multi-omic (CAMO) cohort, part of the Norwegian Women and Cancer (NOWAC) study

**André Berli Delgado**[1]*, **Eline Sol Tylden**[1], **Marko Lukic**[2], **Line Moi**[3], **Lill-Tove Rasmussen Busund**[1,3], **Eiliv Lund**[2], **Karina Standahl Olsen**[2]

1 Department of Medical Biology, Faculty of Health Sciences, UiT - The Arctic University of Norway, Tromsø, Norway, 2 Department of Community Medicine, Faculty of Health Sciences, UiT - The Arctic University of Norway, Tromsø, Norway, 3 Department of Clinical Pathology, University Hospital of North Norway, Tromsø, Norway

* Ade014@uit.no

## Abstract

**Data Availability Statement:** The questionnaire and registry data are stored and managed by the NOWAC research group at the Department of Community Medicine, UiT The Arctic University of Norway, Tromsø, Norway. Blood samples collected

### Introduction

Breast cancer is the most common cancer worldwide and the leading cause of cancer related deaths among women. The high incidence and mortality of breast cancer calls for improved prevention, diagnostics, and treatment, including identification of new prognostic and predictive biomarkers for use in precision medicine.

### Material and methods

With the aim of compiling a cohort amenable to integrative study designs, we collected detailed epidemiological and clinical data, blood samples, and tumor tissue from a subset of participants from the prospective, population-based Norwegian Women and Cancer (NOWAC) study. These study participants were diagnosed with invasive breast cancer in North Norway before 2013 according to the Cancer Registry of Norway and constitute the Clinical and Multi-omic (CAMO) cohort. Prospectively collected questionnaire data on lifestyle and reproductive factors and blood samples were extracted from the NOWAC study, clinical and histopathological data were manually curated from medical records, and archived tumor tissue collected.

### Results

The lifestyle and reproductive characteristics of the study participants in the CAMO cohort (n = 388) were largely similar to those of the breast cancer patients in NOWAC (n = 10 356). The majority of the cancers in the CAMO cohort were tumor grade 2 and of the luminal A subtype. Approx. 80% were estrogen receptor positive, 13% were HER2 positive, and 12% were triple negative breast cancers. Lymph node metastases were present in 31% at diagnosis. The epidemiological dataset in the CAMO cohort is complemented by mRNA, miRNA, and metabolomics analyses in plasma, as well as miRNA profiling in tumor tissue.

as part of the NOWAC Post-genome cohort (plasma, buffy coat, and PAXgene Blood RNA samples) are kept at the UiT Core Facility for Biobanking. The clinical sample material and the clinical database are kept and managed by the Department of Clinical Pathology, University Hospital of North Norway, Tromsø, Norway. Due to the sensitivity of the data that has been collected in the CAMO cohort, both from the data material collected in NOWAC and the medical records as well as from the linkage to the Cancer Registry, the data cannot be placed in a public repository. Data cannot be shared publicly since the data is of a detailed and sensitive nature. However, data will be available upon request to NOWAC at nowac@uit. no.

**Funding:** This study was funded by UiT The Arctic University of Norway, and the NOWAC study was supported by a grant from the European Research Council (ERC-AdG 232997 TICE). The publication charges for this article have been funded by a grant from the publication fund of UiT The Arctic University of Norway. The funders had no role in study design, data collection and analysis, decision to publish, or preparation of the manuscript.

**Competing interests:** The authors have declared that no competing interests exist.

**Abbreviations:** BMI, Body mass index; CAMO, Clinical- and Multi-omic; DPA, Data Protection Authority; ER, Estrogen receptor; FFPE, Formalin-fixed paraffin-embedded; GCF, Genomics Core Facility; HE, Hematoxylin and eosin; HER2, Human epidermal growth factor receptor 2; HRT, Hormone replacement therapy; IHC, Immunohistochemistry; LC, Liquid chromatography; MS, Mass spectrometry; NOWAC, Norwegian Woman and Cancer; OC, Oral contraceptives; PR, Progesterone receptor; SISH, Silver in situ hybridization; TMA, Tissue microarray.

Additionally, histological analyses at the level of proteins and miRNAs in tumor tissue are currently ongoing.

## Conclusion

The CAMO cohort provides data suitable for epidemiological, clinical, molecular, and multi-omics investigations, thereby enabling a systems epidemiology approach to translational breast cancer research.

## Introduction

Breast cancer is diagnosed in more than two million individuals each year and has recently overtaken lung cancer as the most commonly diagnosed cancer worldwide [1]. It is also the leading cause of cancer related deaths among women [2], with more than 680 000 deaths globally in 2020 [3]. The increasing incidence and high mortality of breast cancer call for improved diagnostic biomarkers for early detection, as well as prognostic and predictive biomarkers for precise treatment stratification.

One of the most important factors for reducing morbidity and mortality in breast cancer is early detection, as prognosis strongly depends on the stage of the disease at diagnosis. At present, the 5-year survival of breast cancer in high-income countries is up to 99% in cases of localized disease, but less than 30% if distant metastases are present [4]. This underlines the importance of diagnostic biomarkers for early disease detection.

To account for the heterogeneity between and within tumors, breast cancer treatment is becoming increasingly tailored. The goal of precision medicine is to improve the clinical outcome by detailed knowledge of the disease and targeted treatment, based on the individual's genetic, biomarker, phenotypic, and lifestyle characteristics [5]. Further improvement and personalization of breast cancer treatment is necessary and requires identification of new prognostic and predictive biomarkers to support clinical decision making, as well as novel personalized treatment strategies targeting molecular tumor-specific sites.

Systems science relies on the idea that studying components interacting within a system gives a better understanding of their function and effects than studying each component in isolation [6]. Systems epidemiology enables identification of contributors to disease and their interactions by combining human genomic, transcriptomic, proteomic, and metabolomic data with measurements from observational epidemiologic studies [7]. The complexity of the carcinogenic process, the latency time, and the changing lifestyle of study participants argue for a systems epidemiology approach in cancer research [8].

The large, nationally representative Norwegian Women and Cancer (NOWAC) study is therefore highly interesting since it offers very detailed epidemiological data, also on lifestyle factors, retrieved from repeated questionnaires, as well as blood samples. Here we present our systems epidemiology cohort, the Clinical and Multi-omic (CAMO) cohort, of 388 female patients with invasive breast cancer, from whom we have detailed data retrieved from the questionnaires, blood samples, tissue biopsies, medical records, and the Cancer Registry of Norway. Our cohort provides a unique opportunity to combine several types of data from a wide range of sources, thereby unlocking the potential of a systems epidemiology and precision medicine approach to breast cancer.

## Materials and methods

### Study population

The CAMO cohort includes 388 female breast cancer patients from the Post-genome cohort within the NOWAC study (Fig 1). The NOWAC study is a prospective cohort study, which started in 1991 and included women aged 30–70 years at recruitment, randomly selected from the Norwegian Central Population Register, and irrespective of any previous cancer diagnosis [9]. Baseline information in NOWAC was collected during the years 1991–2007, the first follow-up was conducted in 1998–2014, and the second follow-up in 2004–2011. The NOWAC study data initially comprised detailed questionnaire information on lifestyle, reproductive factors, and medication, including use of hormone replacement therapy (HRT), from more than 172 000 women.

During the years 2003–2006, the NOWAC study was expanded to collect plasma, buffy coat, and blood samples for whole-genome expression profiling using the PAXgene Blood RNA system (Preanalytix/Qiagen, Hilden, Germany). Participants received a blood sampling kit via mail, and blood was drawn at general practitioners' offices. The samples were collected irrespective of any previous breast cancer diagnosis. Isolated RNA is stored in the cohort biobank, enabling future research with the use of updated technology. The resulting NOWAC Post-genome cohort includes 50 000 randomly selected NOWAC participants born in the years 1943–57 [10]. NOWAC participants with a diagnosis of breast cancer were identified through linkage to the Cancer Registry of Norway (update of 2018), using the unique 11-digit personal number assigned to every legal resident in Norway. Information on causes of deaths were acquired from the National Register for Causes of Death, and breast cancer deaths were defined as those with the ICD-10 code 50.

The patients included in the present project are the 388 participants in the NOWAC Post-genome cohort registered in North Norway and diagnosed with breast cancer before 2013. An overview of events such as data collection periods, year of breast cancer diagnosis for the CAMO participants, and changes in clinical practice during the study period is given in Fig 2.

Among the 388 CAMO participants, 313 women (81%) answered the baseline questionnaire before receiving a breast cancer diagnosis (Table 1). Also, 181 (46.8%) of the CAMO participants gave a pre-diagnostic blood sample as part of the NOWAC Post-genome cohort. Conversely, 206 CAMO women were diagnosed with breast cancer after giving a blood sample through the Post-genome cohort.

### Questionnaire data

At inclusion, all NOWAC participants answered a 4-8-page questionnaire regarding health, lifestyle, diet, and reproductive factors. In addition to the baseline information, the majority of the women answered one or two intermediate follow-up questionnaires at 6-8-year intervals. Furthermore, participants in the NOWAC Post-genome cohort answered a 2-page questionnaire accompanying the blood sample collection (Table 1). Questionnaires and blood sampling kits were administered irrespective of any cancer diagnosis. The questionnaires included self-reporting of family history of breast cancer (mother, sister), lifestyle factors such as smoking status (never, former, current), education, body mass index (BMI), physical activity (low, moderate high), alcohol consumption (g/day), and reproductive factors such as reproductive history (number of children, age at first birth), use of oral contraceptives, and use of HRT. For the breast cancer cases in the NOWAC cohort as a whole, we used data from the Cancer Registry to obtain age at diagnosis, and we combined age at diagnosis with menopausal information from NOWAC questionnaires to calculate the variable menopausal status at diagnosis. In the

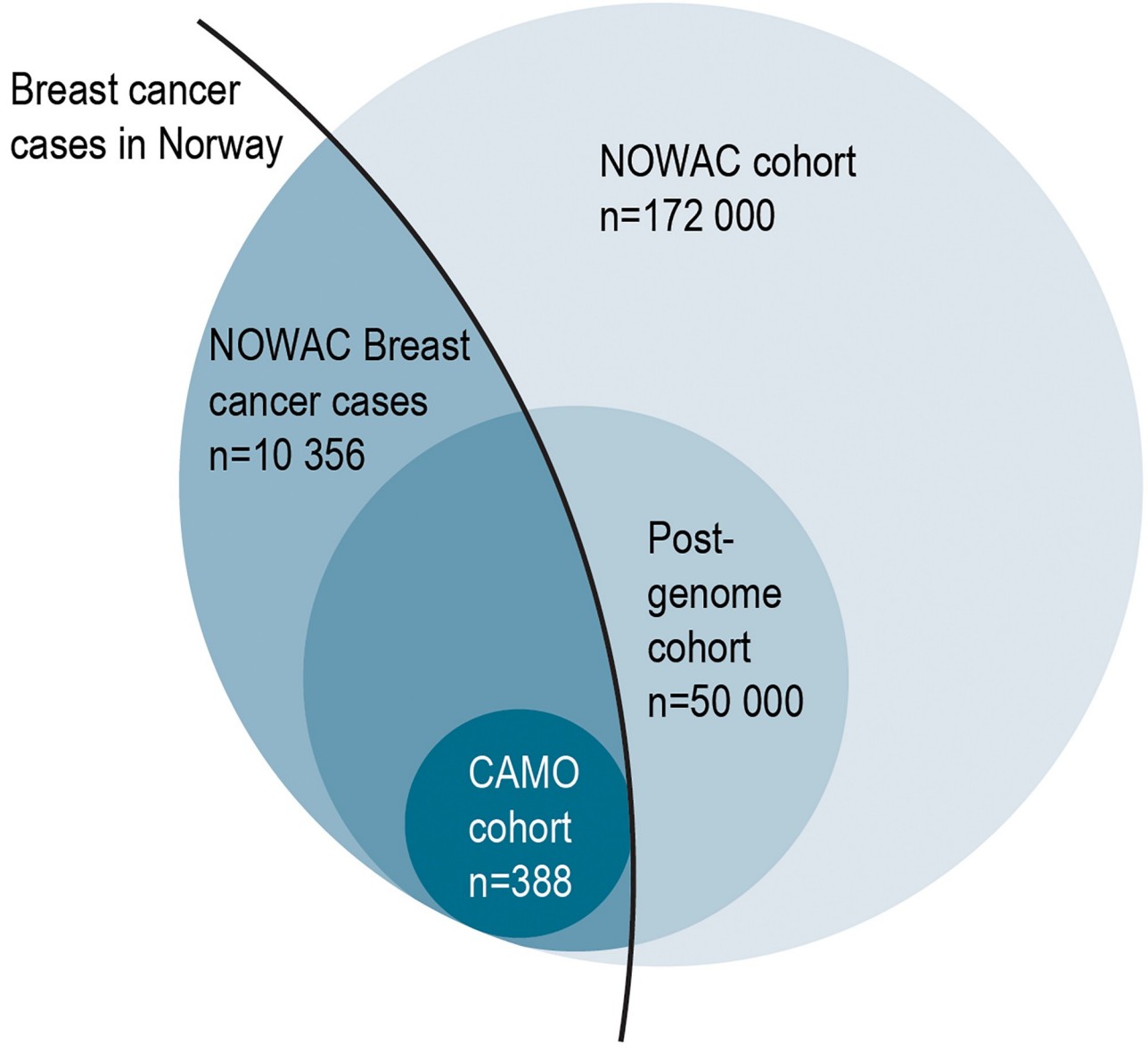

**Fig 1. Study population.** Venn diagram showing size and overlap of study populations in the Norwegian Women and Cancer (NOWAC) study, the NOWAC Post-genome cohort and the Clinical and Multi-omic (CAMO) cohort.

case of missing information, we classified women as pre- or postmenopausal using an age cut-off of 53 years.

## Clinical database

Diagnostic biopsies and breast cancer surgery were performed at the University Hospital of North Norway in Tromsø or at the Nordland Hospital in Bodø. Clinical data were collected through manual review of medical records carried out between the years 2017 and 2019, and included age at diagnosis, treatment modality (endocrine treatment, chemotherapy, anti-HER2, radiation), cancer stage, cancer-related death, distant metastasis, and relapse after

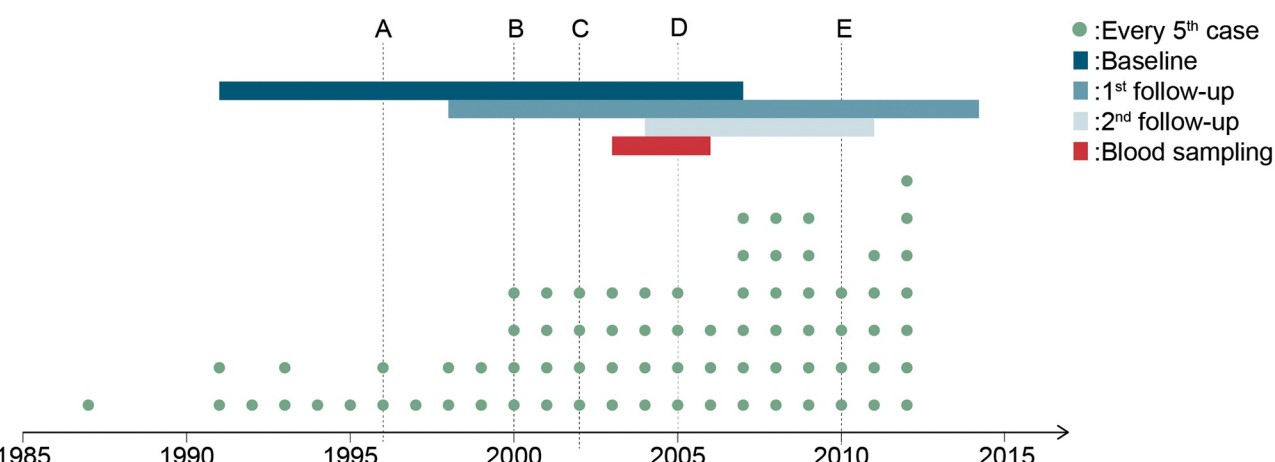

**Fig 2. Timeline of the study period.** Timeline of the study period showing year of breast cancer diagnosis of the study participants (dots), as well as notable changes in screening, diagnostic, and treatment regimes in Norway (vertical lines). Line A denotes the introduction of the national breast cancer (BC) screening program, line B the introduction of chemotherapy regimens AC and FEC in the Norwegian national guidelines for BC treatment, line C the introduction of HER2 analysis in BC diagnostics, line D the use of paclitaxel and docetaxel, aromatase inhibitor, and adjuvant trastuzumab in BC treatment, and line E the introduction of Ki67 analysis and change of ER cutoff to 1% in BC diagnostics. The time periods of data collection in the NOWAC study are shown as colored, horizontal bars. Abbreviations: AC = doxorubicin (also known as Adriamycin) and cyclophosphamide; ER = estrogen receptor; FEC = 5-fluorouracil, epirubicin and cyclophosphamide.

treatment, and histopathological data. All histopathological data, including receptor status, were re-evaluated by a breast pathologist in the study.

Tumor grade was evaluated clinically as part of routine diagnostic assessment, based on gland formation, nuclear pleomorphism, and mitotic count, using the criteria modified by Elston and Ellis [11]. Immunohistochemical (IHC) analyses of estrogen receptor (ER), progesterone receptor (PR) and human epidermal growth factor receptor 2 (HER2) were done on needle biopsies, also as part of routine diagnostic evaluation. Several tumor markers were reanalyzed as part of the present study. To ensure diagnostic quality, we reanalyzed ER and PR for all breast cancers diagnosed before 2001. Cut-off value for ER positivity was $\geq$ 1% and for PR $\geq$ 10% as recommended in the Norwegian national guidelines. HER2 was analyzed as part of this study for all breast cancers diagnosed before HER2-analysis became part of routine practice between 2002 and 2003. Cut-off value for HER2 positivity was an IHC score of 3+, while scores of 0–1+ were considered negative. An IHC score of 2+ led to further assessment of HER2 status by silver *in situ* hybridization (SISH), where a HER2/chromosome 17-ratio $\geq$ 2.0 was considered positive. Since Ki67 was not part of the routine diagnostic protocol before 2011, Ki67 assessment was done as part of the present study, for all luminal cancers diagnosed before 2011. Ki67 expression was measured in histological slides of tumor tissue from the primary surgery to differentiate between luminal A and luminal B tumors. Ki67 expression was evaluated in at least 500 tumor cells in the most proliferative active parts of the tumors and reported as the percentage of positive tumor cell nuclei.

**Table 1. Number of CAMO study participants with pre- or post-diagnostic information from baseline and follow-up questionnaires.**

|  | Total n | Years of sampling | Diagnosed before data collection (n, %) | Diagnosed after data collection (n, %) |
|---|---|---|---|---|
| Baseline | 388 | 1991–2007 | 75 (19.3) | 313 (80.7) |
| 1st follow-up | 354 | 1998–2014 | 155 (43.8) | 199 (56.2) |
| 2nd follow-up | 228 | 2004–2011 | 164 (71.9) | 64 (28.1) |
| Blood sampling | 388 | 2003–2006 | 181 (46.8) | 206 (53.2) |

Molecular subtyping of tumors was done according to recommendations by the St. Gallen International Expert Consensus and previous publications [12, 13], based on the surrogate markers ER, PR, HER2 and Ki67 as follows: luminal A (ER+ and/or PR+, HER2- and Ki67 ≤ 30%), luminal B (ER+ and/or PR+, HER2- and Ki67 > 30% or ER+ and/or PR+ and HER2+), HER2 positive (ER- and PR- and HER2+) and basal-like (ER-, PR- and HER2-).

## Analyses of blood samples

For all 388 women in the clinical cohort, blood samples were collected as part of the NOWAC Post-genome cohort as described previously [10]. In a subset of these samples, i.e. approximately 100 samples, a series of omics analyses have already been carried out. For details on the laboratory methods, please refer to S1 Appendix. Briefly, mRNA gene expression profiles were analyzed from PAXgene blood RNA samples by Illumina HumanHT-12 Expression BeadChip microarrays (Illumina, Inc. San Diego, CA, USA). The analyses were carried out by a certified Illuimna service provider, the Genomics Core Facility (GCF), Norwegian University of Science and Technology, Trondheim, Norway. For plasma metabolomics, a liquid chromatography-mass spectrometry (LC-MS/MS) based kit "AbsoluteIDQ p180" (Biocrates Life Sciences, Innsbruck, Austria) was used for quantification of up to 188 metabolites. The Swedish Metabolomics Centre in Umeå, Sweden carried out the analyses. Extraction and profiling of miRNA in plasma were performed by Exiqon Services (Vedbaek, Denmark). Total RNA was extracted using the mirCURY™ RNA isolation kit, and a PCR-based panel of 372 probes was used for miRNA profiling (miRCURY LNA™ Universal RT microRNA PCR Human panel I, Qiagen, Hilden, Germany). See S1 Appendix for detailed information on the analyses of blood samples.

## Analyses of tumor tissue

Archived formalin-fixed paraffin-embedded (FFPE) tissue blocks were retrieved from the two pathology labs together with the corresponding hematoxylin and eosin (HE) slides.

Tissue microarrays (TMA) were constructed from all available tumor blocks from breast cancer resection specimens from the CAMO cohort participants. The histological slides were evaluated by a pathologist and representative areas of tumor tissue in the invasive front and in the center of the tumor were carefully selected and marked. The TMAs were constructed using a tissue-arraying instrument (Beecher Instruments, Silver Spring, MD) as described in previous publications [14]. In short, a 0.6 mm-diameter stylet was used to collect several replicate tissue cores from each donor block, which were then transferred to a recipient block. Sections of 4 μm were cut with a Microm microtome HM355S (Microm, Walldorf, Germany). TMAs are used for high throughput visualization of molecular targets on DNA-, miRNA-, mRNA- or protein level, using HE staining, IHC, and in situ hybridization (ISH). Assessment of a wide range of potential biomarkers using TMAs is planned or currently ongoing [15].

In a pilot study including 108 of the CAMO participants, we extracted and analyzed l miRNA from FFPE tissue blocks (Exiqon, Vedbaek, Denmark). See S1 Appendix for details. Blood samples from the participants in this pilot study were included in the plasma metabolomics, miRNA analyses and mRNA expression analyses described above.

## Statistical analysis

We compared characteristics of the participants in the CAMO cohort with the rest of the breast cancer cases in the NOWAC study by using independent sample t-test for continuous variables, or chi-square test for categorical variables. The results are presented either as means with standard deviations, medians with data range, or as percentages. All the analyses were

done in STATA version 16.1 (Stata Corp, College Station, TX, USA) and SPSS version 26 (SPSS Inc, Chicago, IL, USA).

### Ethical considerations

The NOWAC study and the study of miRNA expression in tumor samples from the NOWAC Post-genome cohort participants have been approved by the regional ethical committee of North Norway (REKnord 2010/1931, 2013/2271, 2014/1605). The approval covers the collection of lifestyle information, blood samples, cancer tissue, storing of data, and linkage to national registries. In addition, the Norwegian Data Protection Authority (DPA) has approved the storing of all relevant, not identifiable data, and the linkage to national registries. All the participants in NOWAC have given broad written consent to the use of the collected information and biological material for research. The participants can withdraw from the study at any time and can request that the collected samples and information are deleted. Ethical aspects have been considered within the project to ensure the most efficient and accurate use of the collected material, all in accordance with national and international guidelines and laws.

### Results

For the 388 women included in the CAMO cohort, we have compiled information from baseline questionnaires, blood samples, tumor tissue, and a comprehensive clinical database (Fig 3). Median between baseline and 1st follow-up was 5,8 years (range: 4.3–10.0 years), whereas the median between 1st and 2nd follow-up was 8.5 years (range: 5.3–8.8 years).

The mean age at diagnosis of breast cancer in the CAMO cohort is 55.1 years compared to 57.6 years in the NOWAC study as a whole ($p < 0.001$) (Table 2). The percentage of postmenopausal breast cancer cases in the CAMO cohort is 69.1%, which is comparable to 69.3% in the rest of the NOWAC study ($p = 0.91$). There were no significant differences in physical activity and smoking status between the two cohorts; the majority of the participants reported being former smokers and moderately physically active at enrolment ($p = 0.31$ and $p = 0.82$, respectively). Similarly, we found no statistically significant differences in the mean duration of education, number of children, and use of HRT. Compared to rest of the breast cancer cases in the NOWAC cohort, the women in the CAMO cohort have somewhat lower age at first birth (23.8 years vs. 24.4 years, $p = 0.02$), lower age at menarche (13.1 years vs. 13.3 years, $p = 0.002$), higher BMI (24.8 kg/m$^2$ vs. 24.3 kg/m$^2$, $p = 0.03$), reported drinking less alcohol (3.3 g/day vs. 4.2 g/day, $p < 0.001$), and were less likely to report a positive history of breast cancer in mothers (5.2% vs. 9.5%, $p = 0.006$) and sisters (2.1% vs. 5.3%, $p = 0.01$). Finally, 63.2% of all deaths that occurred during follow-up in the CAMO cohort were breast cancer related, compared to 61.8% in the NOWAC cohort, however, the difference was not significant ($p = 0.83$).

Histological grade 2 was the most frequent tumor grade, observed in 42.5% of the breast tumors, and the median tumor diameter was 15 mm (Table 3). The most frequent molecular subtypes were luminal A (58.0%) followed by luminal B (19.9%). More women underwent lumpectomy (67.3%) compared to primary mastectomy (32.2%), and one participant did not undergo any surgery. At diagnosis, most women did not have lymph node metastases (68.6%). The number of lymph node metastases at diagnosis ranged from 0 to 32, with a median value of zero. Up until the data collection for the clinical database started in 2017, only 10.8% of cancers had metastasized to distant sites, whereas 8% of cancers had relapsed. Most tumors were hormone receptor positive: 81.3% were ER positive and 62.6% were PR positive. 12.9% of breast cancers were HER2 positive. Radiation was the most common adjuvant therapy (73.7%), followed by anti-estrogen (52.8%) and chemotherapy (37.9%). Only 6.4% of the

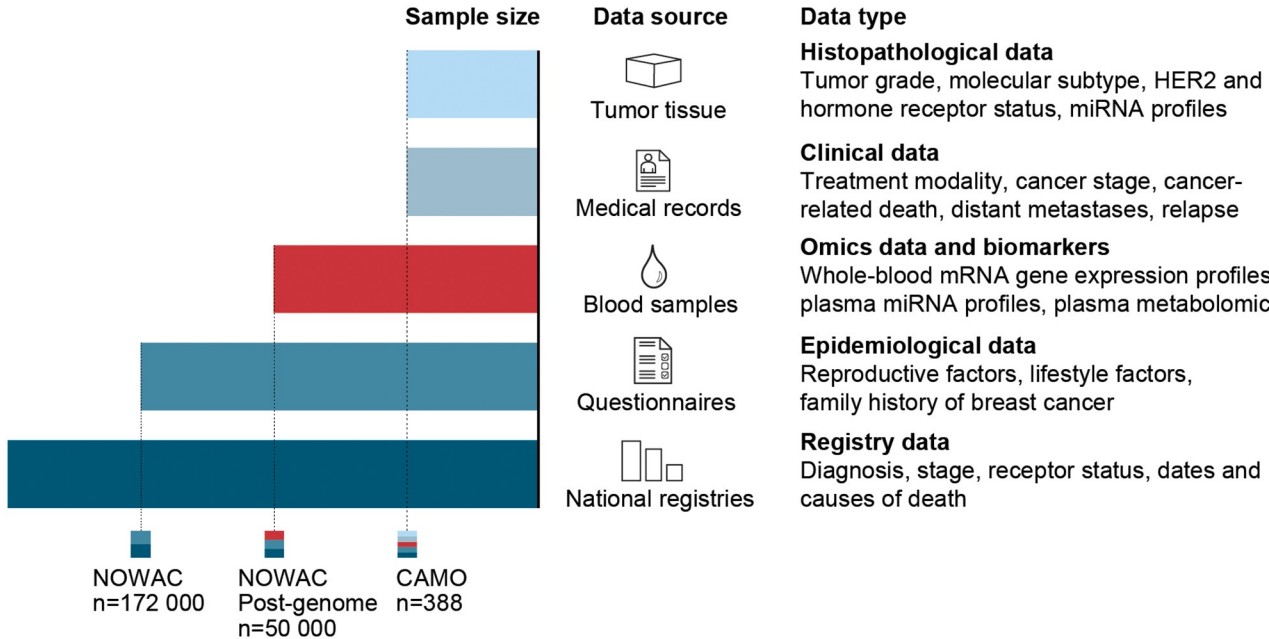

**Fig 3. Overview of sample sizes, data sources, and data types.** The Clinical and Multi-omic (CAMO) cohort, nested within the Norwegian Women and Cancer (NOWAC) study, provides multiple types of data from a wide range of sources, thereby enabling a systems epidemiology approach to breast cancer. These multimodal data may be combined in various ways to create complex study designs that can be used to investigate hypotheses related to breast cancer prevention, diagnostics, treatment, and survival.

participants received anti-HER2 therapy. We observed the highest number of missing values in the following variables: molecular subgroup (5.7%), PR status (4.9%), and HER2 status (4.6%).

## Discussion

Nested within the NOWAC cohort, the CAMO cohort of 388 female breast cancer patients lays the foundation for integrative study designs. By merging epidemiological, clinical, molecular, and multi-omics data retrieved from national registries, questionnaires, medical records, blood samples and tissue biopsies, the CAMO cohort enables a systems epidemiology approach to breast cancer.

The CAMO cohort includes a set of already analyzed data for several molecular markers and omics profiles, from both blood samples and tumors. Molecular markers in tumors have been analyzed for the entire cohort, whereas a core of approximately 100 samples is approaching the full systems epidemiology potential. In this core subsample, the following omics data is available, in addition to the molecular tumor markers: whole-blood mRNA gene expression profiles, miRNA profiles and metabolomics in plasma, and miRNA profiles in tumors.

To date, our published findings on these 100 samples have been based on the tumor material and have focused on single markers or profiles. We identified significantly different miRNA profiles between malignant and normal breast tissue, and between cancer subgroups according to ER status, tumor grade and molecular subtype [16]. miRNAs in the miR-17-92 cluster and miR-17 family were overexpressed in high grade and triple-negative tumors. Further, we showed that the expression of miR-143 and miR-145 was lower in malignant compared to normal breast tissue, and lower in the more aggressive tumors with higher tumor grade, loss of ER and the basal-like phenotype [17].

**Table 2. Comparison of selected characteristics between the study populations of CAMO (n = 388) and NOWAC breast cancer cases (n = 10 356).**

| Characteristics at study enrolment | CAMO cohort | NOWAC breast cancer cases | p-value |
|---|---|---|---|
| Participants at baseline, n (%) | 388 | 10 356 | |
| Age at diagnosis (y), mean (SD) | 55.1 (7.0) | 57.6 (9.7) | <0.001 |
| Smoking status at baseline, n (%) | | | |
| Never | 128 (33.2) | 3 435 (33.6) | 0.82 |
| Former | 146 (37.8) | 3 710 (36.3) | |
| Current | 112 (29.0) | 3 078 (30.1) | |
| Duration of education (y), mean (SD) | 12.3 (3.5) | 12.5 (3.5) | 0.23 |
| Body mass index, mean (SD) | 24.8 (4.7) | 24.3 (3.8) | 0.03 |
| Physical activity level, n (%) | | | |
| Low | 98 (28.1) | 2 549 (26.8) | 0.31 |
| Moderate | 155 (44.4) | 3 987 (41.9) | |
| High | 96 (27.5) | 2 986 (31.4) | |
| Alcohol consumption (g/day), mean (SD) | 3.3 (4.5) | 4.2 (5.7) | <0.001 |
| Number of children, mean (SD) | 2.2 (1.1) | 2.1 (1.2) | 0.12 |
| Age at first birth (y), mean (SD) | 23.8 (4.6) | 24.4 (4.6) | 0.02 |
| Age at menarche (y), mean (SD) | 13.1 (1.3) | 13.3 (1.4) | 0.002 |
| Ever use of oral contraceptives at baseline, n (%) | 229 (61.1) | 5 648 (56.7) | 0.09 |
| Use of hormone replacement therapy at baseline, n (%) | | | |
| Never | 223 (65.8) | 3 729 (62.8) | 0.063 |
| Former | 31 (9.1) | 809 (13.6) | |
| Current | 85 (25.1) | 1 405 (23.6) | |
| Maternal history of breast cancer, n (%) | 19 (5.2) | 894 (9.5) | 0.006 |
| Sister history of breast cancer, n (%) | 7 (2.1) | 407 (5.3) | 0.01 |
| Menopausal status at diagnosis, n (%) | | | |
| Pre | 120 (30.9) | 3 186 (30.7) | 0.91 |
| Post | 268 (69.1) | 7 201 (69.3) | |
| Cause of death, n (%) | | | |
| Breast cancer | 36 (63.2) | 1222 (61.8) | 0.83 |
| Non-breast cancer | 21(36.8) | 757 (38.3) | |

SD: standard deviation

From the blood sampling in the NOWAC Post-genome cohort, single-omics data have been published in combination with use of data from the National Cancer Registry. In prospective analyses, we identified time-dependent changes of the blood transcriptome up to 8 years before breast cancer diagnosis [18], and found potentially wide-reaching differences in blood gene expression profiles between metastatic and non-metastatic breast cancer cases up to two years before diagnosis [19]. In blood samples taken after the breast cancer diagnosis, a transient increase in the number of differentially expressed genes was identified at 3–4 years after diagnosis, but only in patients who later died [20]. Along the same lines, we provided a proof of concept for the use of blood gene expression profiles as biomarkers of death from metastatic breast cancer [21].

We have explored associations of blood gene expression profiles and several breast cancer related lifestyle and dietary exposures. For example, we showed that each pregnancy changes blood gene expression profiles in a linear fashion relative to the number of children. However, this was only found in healthy women, and not in women who later developed breast cancer

**Table 3. Clinical characteristics of the participants in the CAMO cohort (n = 388).**

| Characteristics | Estimate | Missing, n (%) |
|---|---|---|
| Tumor diameter in millimeters, median (range) | 15 (0.1–92) | 3 (0.8) |
| Tumor grade, n (%) | | 7 (1.8) |
| 1 | 121 (31.2) | |
| 2 | 165 (42.5) | |
| 3 | 95 (24.5) | |
| Molecular subgroup, n (%) | | 22 (5.7) |
| Luminal A | 225 (58.0) | |
| Luminal B | 77 (19.9) | |
| HER2+ | 18 (4.6) | |
| Basal-like | 46 (11.9) | |
| Surgery, n (%) | | 1 (0.3) |
| None | 1 (0.3) | |
| Lumpectomy | 261 (67.3) | |
| Mastectomy | 125 (32.2) | |
| Lymph node metastasis, n (%) | | 3 (0.8) |
| Yes | 119 (30.7) | |
| No | 266 (68.6) | |
| Number of lymph node metastases, median (range) | 0 (0–32) | 4 (1.0) |
| Distant metastasis*, n (%) | | 3 (0.8) |
| Yes | 42 (10.8) | |
| No | 343 (88.4) | |
| Relapse, n (%) | | 2 (0.5) |
| Yes | 31 (8.0) | |
| No | 355 (91.5) | |
| ER positive, n (%) | | 3 (0.8) |
| Yes | 313 (81.3) | |
| No | 72 (18.7) | |
| PR positive, n (%) | | 19 (4.9) |
| Yes | 243 (62.6) | |
| No | 126 (32.5) | |
| HER2 positive, n (%) | | 18 (4.6) |
| Yes | 50 (12.9) | |
| No | 320 (82.5) | |
| Adjuvant treatment, n (%) | | |
| Radiation | 286 (73.7) | 7 (1.8) |
| Chemotherapy | 147 (37.9) | 9 (2.3) |
| Anti-estrogen | 205 (52.8) | 8 (2.1) |
| Anti-HER2 | 25 (6.4) | 5 (1.3) |

ER: estrogen receptor, HER2: human epidermal growth factor receptor 2, PR: progesterone receptor,

SD: standard deviation

*Distant metastasis at any time point

[22]. Several other lifestyle and dietary factors also impact blood gene expression, ultimately affecting the molecular and immunological processes potentially involved in breast cancer. These include smoking [23], HRT [24], dietary fatty acids [25], vitamin D [26] and coffee consumption [27].

Collectively, the studies published to date based on either the CAMO or the NOWAC Post-genome cohort demonstrate the potential of the available data to provide new insight into cancer development and progression, and its potential to contribute to precision medicine. We have demonstrated the importance of incorporating questionnaire data with molecular markers and profiles, and the importance of the clinical data for patient stratification. Also, the time factor is essential when studying the trajectories of molecular profiles before, at, and after diagnosis. In CAMO, serial sampling of biological specimens was not carried out. To account for this, we have approached the time issue using group level data, exploiting the randomness of the length of follow-up time due to the study design of the NOWAC Post-genome cohort.

Future use of the data from the CAMO cohort will move into multi-level analyses, by incorporating data from both blood and tumor with data from questionnaires, whole slide images, lymph node analyses, multiple omics profiles, and molecular markers, all measured at different time points in the trajectory going from the healthy to the disease state. These data will be combined with information from the clinical database, and end point registries (Fig 3). The multiple dimensions that can be combined for each possible study design has been described elsewhere [28]. Briefly, the combination of the following dimensions will create a multitude of possible study designs: time, exposures, measurements, diagnosis, participant selection, sample types, as well as stratification and de-confounding. The process of designing each study will be stepwise, by defining content to each dimension, and making choices relevant to the study question at hand. Several challenges accompany each dimension of this systems epidemiology and precision medicine approach. Challenges include the risk of systematic errors, the adequate collection and preservation of biological samples, building computer system architectures for data management and analysis, as well as the need for advanced analytical approaches for dimension reduction, blockwise missingness of data, data pattern exploration, and causal inference.

The main strengths of the CAMO cohort include its prospective design, the relatively large sample size compared to other similar cohorts, the large number of variables available from the clinical database, and the NOWAC questionnaires. The use of follow-up questionnaires allows research questions aiming to explore associations between pre- and post-diagnostic factors and clinical outcomes, analyzing effects of changes in exposures, and attenuates the risk of measurement error. The NOWAC study was previously externally validated by Lund et al. 2003, revealing no major source of selection bias [29]. The results presented here indicate that there are no substantial differences between the CAMO cohort and the rest of the breast cancer cases in the NOWAC study, other than alcohol consumption and family history of breast cancer. Although we found statistically significant differences in other characteristics such as age at diagnosis, BMI, age at first birth, and age at menarche, these differences are clinically negligible. This, along with the fact that all breast cancer cases from the NOWAC Post-genome cohort living in North Norway were the basis of inclusion in the CAMO cohort, ensures high external validity.

Due to the size of the CAMO cohort, we might lack statistical power to assess associations of less frequent biomarkers, exposures, and outcomes. A certain degree of misclassification is present due to the self-reporting of questionnaire data. For some of the women in our cohort, a large gap between the time of questionnaire data collection and the point of diagnosis may also contribute to misclassification. Finally, data collection from clinical and medical databases and records poses several issues. Despite best efforts, this type of data extraction is non-automatic, and prone to technical and human error due to the handling of very large data sets. Other data sources for verification and validation of the extracted data are not available.

## Conclusion

The availability of biological samples from study participants is a prerequisite in systems epidemiological studies. Experiments using tissue from study participants enable investigation of molecular marker profiles resulting from complex real-life situations, potentially revealing how various exposures may affect disease characteristics, development, and progression. The CAMO cohort, nested within the NOWAC study, provides multi-dimensional data from a wide range of sources. These data may be combined in various ways to create a multitude of complex study designs, allowing investigation of a great variety of hypotheses related to breast cancer prevention, diagnostics, treatment, and survival. Using a systems epidemiology approach, we may increase our understanding of biological pathways in breast cancer as time- and exposure dependent trajectories and investigate how these trajectories differ by clinical strata. This, in turn, lays the groundwork for discovery of new diagnostic, prognostic and predictive biomarkers that may facilitate early diagnosis, support clinical decision-making, and improve the precision of interventions.

## Supporting information

**S1 Appendix. Analyses of blood samples and tumor tissue.**
(DOCX)

## Acknowledgments

We are grateful to the NOWAC participants for contributing their information, blood, and tissue samples. We also acknowledge the invaluable help from technical and administrative staff in NOWAC, and from Kari Wagelid Grønn who contributed with graphic design of the figures.

   **Disclaimer:** Some of the data in this work are from The Cancer Registry of Norway. The Cancer Registry of Norway is not responsible for the analysis or interpretation of the data.

## Author Contributions

**Conceptualization:** Lill-Tove Rasmussen Busund, Eiliv Lund, Karina Standahl Olsen.

**Data curation:** André Berli Delgado, Eline Sol Tylden, Line Moi, Karina Standahl Olsen.

**Formal analysis:** André Berli Delgado, Eline Sol Tylden, Marko Lukic.

**Funding acquisition:** André Berli Delgado, Eline Sol Tylden.

**Investigation:** André Berli Delgado, Eline Sol Tylden, Marko Lukic, Line Moi, Karina Standahl Olsen.

**Methodology:** André Berli Delgado, Eline Sol Tylden, Marko Lukic, Line Moi, Lill-Tove Rasmussen Busund, Karina Standahl Olsen.

**Project administration:** Lill-Tove Rasmussen Busund, Karina Standahl Olsen.

**Supervision:** Marko Lukic, Line Moi, Lill-Tove Rasmussen Busund, Karina Standahl Olsen.

**Validation:** Marko Lukic, Line Moi, Karina Standahl Olsen.

**Visualization:** Eline Sol Tylden, Karina Standahl Olsen.

**Writing – original draft:** André Berli Delgado, Eline Sol Tylden, Marko Lukic, Line Moi, Karina Standahl Olsen.

**Writing – review & editing:** André Berli Delgado, Eline Sol Tylden, Marko Lukic, Line Moi, Lill-Tove Rasmussen Busund, Eiliv Lund, Karina Standahl Olsen.

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
