## [Decision Letter · Decision Letter 0]

22 Nov 2022

PONE-D-22-25688Cohort profile: The Clinical and Multi-omic (CAMO) cohort, part of the Norwegian Women and Cancer (NOWAC) studyPLOS ONE

Dear Dr. Berli Delgado,

Thank you for submitting your manuscript to PLOS ONE. After careful consideration, we feel that it has merit but does not fully meet PLOS ONE’s publication criteria as it currently stands. Therefore, we invite you to submit a revised version of the manuscript that addresses the points raised during the review process.

We look forward to receiving your revised manuscript.

Kind regards,

Alvaro Galli

Academic Editor

PLOS ONE

Journal Requirements:

"We are grateful to the NOWAC participants for contributing their information, blood, and tissue samples. We also acknowledge the invaluable help from technical and administrative staff in NOWAC, and from Kari Wagelid Grønn who contributed with graphic design of the figures."

"This study was funded by UiT The Arctic University of Norway, and the NOWAC study was supported by a grant from the European Research Council (ERC-AdG 232997 TICE). The publication charges for this article have been funded by a grant from the publication fund of UiT The Arctic University of Norway. The funders had no role in study design, data collection and analysis, decision to publish, or preparation of the manuscript."

Reviewers' comments:

Reviewer's Responses to Questions

**Comments to the Author**

1. Is the manuscript technically sound, and do the data support the conclusions?

Reviewer #1: Yes

Reviewer #2: No

Reviewer #3: Partly

2. Has the statistical analysis been performed appropriately and rigorously? 

Reviewer #1: Yes

Reviewer #2: N/A

Reviewer #3: Yes

3. Have the authors made all data underlying the findings in their manuscript fully available?

Reviewer #1: Yes

Reviewer #2: No

Reviewer #3: No

4. Is the manuscript presented in an intelligible fashion and written in standard English?

Reviewer #1: Yes

Reviewer #2: Yes

Reviewer #3: Yes

5. Review Comments to the Author

Reviewer #1: The authors describe a resource they have developed nested within the Norwegian Women and Cancer (NOWAC) study. The subset described is of 388 breast cancer patients on whom epidemiologic, genomic and breast cancer pathology and treatment data have been gathered. This will be a rich resource for performing "multi-level analyses" at different time points in healthy to disease states to increase understanding of breast cancer development and progression.

Some minor comments:

Why was >10% used as threshold for PR positivity?

Would the data be altered if the current HER2 ratio of 2.0 be used for HER2 amplification?

Is it correct that the dates of the first follow up are 1998-2014? (vs. 1998-2004)?

Given the relatively small size of the cohort (in particular of HER2 enriched and basal-like subtypes), it may be that "support of clinical decision-making" and "precision interventions" is rather more aspirational, and that the study set will be better utilized for hypothesis generation for these types of management goals in women with luminal A-like breast cancers.

Reviewer #2: Delgado and co-workers present a kind of review article that describes some clinical and molecular features of the CAMO patient cohort, and some results the authors had obtained using that cohort. CAMO is part of the Norwegian Women and Cancer study (NOWAC). However, while the former covers just 388 patients, NOWAC has over 10,000 participants.

In the manuscript, some clinical features of the two cohorts are compared and most of them are not substantially different. The authors list a number of mostly own publications that have been published using the CAMO data. Unfortunately, no information is provided how others might make use of that resource (Data availability is annotated as: No – some restrictions will apply). In lines 235-237 the authors write that ‘These multimodal data may be combined in various ways to create complex study designs that can be used to investigate hypotheses related to breast cancer prevention, diagnostics, treatment, and survival’. While this statement may hold, readers would not be able to realize these promises unless they had access to the data. The utility of the CAMO data and of the manuscript are thus limited, particularly along the envisioned lines of epidemiological, clinical, molecular, and multi-omics investigations. Its use towards systems epidemiology in translational breast cancer research seems restricted to the authors and, potentially, their direct collaborators. At least 12 of the 28 references are self-citations (mostly author Lund).

The number of 388 cases in the CAMO appears to be low, given the complexity of breast cancer (subtypes) and of other parameters that are described in the manuscript. Having recruited patients into the study over a period of many years suggests that treatment regimen – intentionally affecting patient outcome – have changed and improved outcome (compare Figure 2). Data from TNBC patients having been treated, for instance with paclitaxel or platinum drugs should be hard to combine with that of patients having received other regimen in earlier years. These issues might impact significance of potential findings. It remains unclear from the manuscript why only a small subset of the NOWAC cohort was selected for CAMO, thereby not leveraging the full potential NOWAC likely has.

Other comments.

Some of the technologies used to collect data are not up-to-date. For example, gene expression profiling data had been collected using Illumina array technology that was discontinued some years ago. It is unclear if that data could be combined with state-of-the-art sequencing data.

In lines 202 and following, the generation of tissue microarrays is described. However, it is not clear how these TMAs have been used and no new findings are described. Instead, the authors write that an assessment of a wide range of potential biomarkers using TMAs is planned or currently ongoing (lines 215-216). Description of these TMAs would be most useful if data was presented having been collected using these tools.

The authors performed some statistical analysis of parameters in NOWAC vs. CAMO studies. For example, they found that the mean age was significantly different. While that in CAMO was 55.1 years, it was 57.6 in NOWAC (p<0.001). Significant differences were found also for e.g., the age at first birth (23.8 vs. 24.4 years, p=0.02), age at menarche, and reported drinking of alcohol. While these differences may be significant, are they also relevant? The numbers are reported in the text and repeated in the Table 2.

In lines 288 and following, some molecular analysis is described, however, no data is presented on molecular markers that would have been identified in a core of approximately 100 samples that is stated to approach the full systems epidemiological potential. The description of novel data and its relevance in breast cancer is absent.

This is a study of limited interest.

Reviewer #3: The manuscript “Cohort profile: The Clinical and Multi-omic (CAMO) cohort, part of the Norwegian Women and Cancer (NOWAC) study” is basically a description of study design, data collection, and representativeness of the breast cancer cases of CAMO compared to the larger NOWAC study. The data collection is surely very unique .

My suggestions to improve readability and clarity of the manuscript, and to focus on what is really the aim of this manuscript are the following (in order of the manuscript):

In the manuscript the CAMO cohort includes women with breast cancer, please clarify if this is only invasive or also DCIS.

Were all women included before or after a first primary breast cancer? How was dealt with women that had another cancer before their breast cancer or were there none?

The figures are quite informative. However, Figure 2 was not immediately clear to me. It would help to add the information related to A-E directly in text boxes with the figure.

Is each dot 5 women? Then I think I only count 380 women?

It is also not visible in the figure which proportion of women had second follow-up is it possible to somehow visualize this? I would otherwise refer to table 1 (see next comment).

In table 1 add the median (range) time between baseline and first and second follow-up. This information is especially needed given the claims in the discussion about the value of having a cohort with different times between cancer-free and after cancer diagnosis.

Page 6 line 129/130. It is a pity that less than half of the women provided a blood sample before breast cancer diagnosis, more incident breast cancers would have been valuable. How does this proportion of incident and prevalent breast cancer (in relation to the blood sampling) in CAMO relates to NOWAC?

Page 7 line 145 “self-reporting of family history of breast cancer (mother, sister),”

Please clarify. Was indeed only family history of mothers and sister reported, not of all first-degree relatives?

Page 8 “To ensure diagnostic quality, we reanalyzed ER and PR for all breast cancers diagnosed before 2001”

Which proportion of the data does this concern? There will also still be substantial variation between hospitals/ laboratories, so why did the authors not re-analyze receptors for all samples? For those that were not stained again, were these re-scored by one pathologist?

Page 14 discussion: “Molecular markers in tumors have been analyzed for the entire cohort, whereas a core of approximately 100 samples is approaching the full systems epidemiology potential.”

Where can this number be found back in the text or tables? Figure 3 and tables 1-3 suggest that all data is available for almost all breast cancer cases. Does this only refer to the part on which there is also omics data? I strongly suggest including a table with omics information in the results section (not necessarily results of analyses, but at least showing which data is available for how many samples). It would also be informative to show how data availability is distributed over core breast cancer subgroups e.g. pre- versus post-menopausal and subtypes (at least ER positive versus negative).

The discussion is overall quite long, would consider reducing the parts on own previous results and on multiple dimensions. Related to previous results, a burning question would be how the CAMO dataset can help with the next steps, this is not really addressed.

One of the most important pieces of information, which is basically table 2 only appears late in the discussion (p17 row 350-356), suggest moving this earlier in the discussion.

The data collection is very unique, but still the sample set is small for multilevel analyses. Perhaps the discussion should be tuned down a bit on the promises for the future.

“In CAMO, serial sampling of biological specimens was not carried out. To account for this, we have approached the time issue using group level data, exploiting the randomness of the length of follow-up time due to the study design of the NOWAC Post genome cohort.”

Indeed, the lack of serial sampling may be mentioned as a shortcoming. However, the current manuscript does not provide any support or evidence in the shown tables or figures that this short coming is covered by the study design and sampling over a longer period. Frankly I would debate that this is possible unless the study would be very large and the distribution of patient and tumor characteristics well balanced over time between samples and diagnosis of cancer.

“The NOWAC study was previously externally validated (28).” Please explain better, not clear what is meant here.

Appendix. Correct typo page 1 “as as”

The appendix explains the scoring of the TMAs but not of the original and newly stained whole slides for ER, PR, HER2.

This sentence in the Appendix “Each researcher gives three scores to each core, reflecting staining density and/or intensity in stroma, tumor cytoplasm and tumor nucleus.” needs some rewriting. I guess not all markers stain in all areas (stroma, cytoplasm, nucleas) and is always either density (is this % of cells?) or intensity (is this vague to strong coloring?) scored?

A better explanation of how the data can be requested, even if restrictions apply, should be included in the data availability statement.

6. PLOS authors have the option to publish the peer review history of their article (what does this mean?). If published, this will include your full peer review and any attached files.

Reviewer #1: No

Reviewer #2: **Yes: **Stefan Wiemann

Reviewer #3: No

---

## [Author Response · Author response to Decision Letter 0]

9 Jan 2023

Issues raised by the Editor

Response: Thank you for pointing this out, files have been updated according to the style requirements.

"We are grateful to the NOWAC participants for contributing their information, blood, and tissue samples. We also acknowledge the invaluable help from technical and administrative staff in NOWAC, and from Kari Wagelid Grønn who contributed with graphic design of the figures."

"This study was funded by UiT The Arctic University of Norway, and the NOWAC study was supported by a grant from the European Research Council (ERC-AdG 232997 TICE). The publication charges for this article have been funded by a grant from the publication fund of UiT The Arctic University of Norway. The funders had no role in study design, data collection and analysis, decision to publish, or preparation of the manuscript."

Response: We have reviewed the manuscript, acknowledgements and funding statement, and we cannot identify any mistakes. Neither the Acknowledgments nor any other parts of the manuscript include any statement regarding funding.

Response: Thank you for pointing out this lack of precision, we have now added the required information in our Ethics statement in the online submission. We have also added the ethics statement in the manuscript under the “Material and methods”-section, line 231-243. 

Reviewer 1

The authors describe a resource they have developed nested within the Norwegian Women and Cancer (NOWAC) study. The subset described is of 388 breast cancer patients on whom epidemiologic, genomic and breast cancer pathology and treatment data have been gathered. This will be a rich resource for performing "multi-level analyses" at different time points in healthy to disease states to increase understanding of breast cancer development and progression. Some minor comments:

Response: We highly appreciate the positive comment from Reviewer 1.

“Why was >10% used as threshold for PR positivity?”

Response: Progesterone receptor analyses have been performed according to national Norwegian guidelines. As recommended in national guidelines, progesterone receptors are analyzed by immunohistochemistry in FFPE tissue biopsies at the time of diagnosis and ≥ 10 % used as threshold for positivity. We have included the phrase “..as recommended in the Norwegian national guidelines” on p. 8, line 171, to clarify this issue in the text.

 “Would the data be altered if the current HER2 ratio of 2.0 be used for HER2 amplification?”

Response: We highly appreciate the reviewer’s attention to important details regarding classification of the tumors. The results of all HER2 in situ hybridization (SISH) have been checked, and the classification would not be altered due to the change in HER2/chromosome 17-ratio cut-off from 2,2 to 2,0. We have updated the manuscript p. 8, line 175-176, to «…HER2/chromosome 17-ratio ≥ 2.0» which is also in line with the current national guidelines regarding HER2-evaluation in breast cancer.

“Is it correct that the dates of the first follow up are 1998-2014? (vs. 1998-2004)?”

Response: Yes, the first follow-up was carried out in waves during the years 1998-2014. For example, some women enrolled in 1991 were invited to a first follow-up in 1998, whereas some women enrolled 2004 were invited for their follow-up in 2014.

“Given the relatively small size of the cohort (in particular of HER2 enriched and basal-like subtypes), it may be that "support of clinical decision-making" and "precision interventions" is rather more aspirational, and that the study set will be better utilized for hypothesis generation for these types of management goals in women with luminal A-like breast cancers.”

Response: We appreciate and agree with the reviewer’s comment. However, further studies using the CAMO cohort can generate hypothesis that can be tested by larger cohorts, which could support clinical decision-making, also in smaller cancer subgroups. 

Reviewer 2

Delgado and co-workers present a kind of review article that describes some clinical and molecular features of the CAMO patient cohort, and some results the authors had obtained using that cohort. CAMO is part of the Norwegian Women and Cancer study (NOWAC). However, while the former covers just 388 patients, NOWAC has over 10,000 participants.

In the manuscript, some clinical features of the two cohorts are compared and most of them are not substantially different. The authors list a number of mostly own publications that have been published using the CAMO data. Unfortunately, no information is provided how others might make use of that resource (Data availability is annotated as: No – some restrictions will apply). In lines 235-237 the authors write that ‘These multimodal data may be combined in various ways to create complex study designs that can be used to investigate hypotheses related to breast cancer prevention, diagnostics, treatment, and survival’. While this statement may hold, readers would not be able to realize these promises unless they had access to the data. The utility of the CAMO data and of the manuscript are thus limited, particularly along the envisioned lines of epidemiological, clinical, molecular, and multi-omics investigations. Its use towards systems epidemiology in translational breast cancer research seems restricted to the authors and, potentially, their direct collaborators. At least 12 of the 28 references are self-citations (mostly author Lund).

Response: We respectfully acknowledge the viewpoint of the reviewer; however, we would like to take the opportunity to highlight the hallmarks of a cohort profile-type paper. Cohort profile papers are designed to fill the space between a study protocol and a research paper. The main motivation of a cohort profile paper is to describe the rationale for assembling the cohort, what methods have been used, the baseline data describing the cohort, and its future plans. We aimed for a very clear referral to such papers, as “cohort profile” is a part of our title. We would argue that such papers are within the scope of PLOS ONE, which we have selected as our primary journal choice due to its broad readership from both the cancer research and computational/statistical modeling side. 

The number of 388 cases in the CAMO appears to be low, given the complexity of breast cancer (subtypes) and of other parameters that are described in the manuscript. Having recruited patients into the study over a period of many years suggests that treatment regimen – intentionally affecting patient outcome – have changed and improved outcome (compare Figure 2). Data from TNBC patients having been treated, for instance with paclitaxel or platinum drugs should be hard to combine with that of patients having received other regimen in earlier years. These issues might impact significance of potential findings. It remains unclear from the manuscript why only a small subset of the NOWAC cohort was selected for CAMO, thereby not leveraging the full potential NOWAC likely has.

Response: We appreciate the reviewer pointing out this limitation. In the CAMO-cohort we have registered data on treatment type, both surgery, radiation therapy, endocrine therapy and chemotherapy. We have registered both which types of chemotherapy the patients have received, as well as the duration of the treatment. 

The subset of 388 patients were chosen because these patients were diagnosed in the same health region in northern Norway. The tumor material where stored in two collaborating departments of clinical pathology, and where not used in any other research projects and therefore available and could be included in the CAMO cohort. Both hospitals, where the tumor material was stored, also use the same system for medical records which enabled us to carry out a thorough review of the records. 

Other comments. Some of the technologies used to collect data are not up-to-date. For example, gene expression profiling data had been collected using Illumina array technology that was discontinued some years ago. It is unclear if that data could be combined with state-of-the-art sequencing data.

Response: It is certainly true that the bead array technology has been surpassed by sequencing technology for the majority of research purposes. Importantly, we do have isolated RNA in the cohort biobank, so that future research can make use of updated technology. We have added a sentence to highlight this in line 104-105.

In lines 202 and following, the generation of tissue microarrays is described. However, it is not clear how these TMAs have been used and no new findings are described. Instead, the authors write that an assessment of a wide range of potential biomarkers using TMAs is planned or currently ongoing (lines 215-216). Description of these TMAs would be most useful if data was presented having been collected using these tools.

Response: We appreciate the importance of presenting data regarding studies using the TMAs, as highlighted by the reviewer. As stated in the manuscript, this work is ongoing, and it is our aim that this manuscript presenting the cohort of breast cancers included in the TMAs would offer a detailed and accurate description of the cohort to future readers and potential collaborators. One article has been published using TMAs from the CAMO-cohort as part of a study on serglycin in breast cancer and is now referred to in the manuscript on p. 10, line 219 as an example of how the TMAs can be used. 

The authors performed some statistical analysis of parameters in NOWAC vs. CAMO studies. For example, they found that the mean age was significantly different. While that in CAMO was 55.1 years, it was 57.6 in NOWAC (p<0.001). Significant differences were found also for e.g., the age at first birth (23.8 vs. 24.4 years, p=0.02), age at menarche, and reported drinking of alcohol. While these differences may be significant, are they also relevant? The numbers are reported in the text and repeated in the Table 2.

Response: We agree with the reviewer that these differences are not necessarily relevant for the cohort or future studies. We list these differences in the text and repeat them in Table 2 to specify that we have considered the potential differences between the CAMO-cohort and NOWAC. 

In lines 288 and following, some molecular analysis is described, however, no data is presented on molecular markers that would have been identified in a core of approximately 100 samples that is stated to approach the full systems epidemiological potential. The description of novel data and its relevance in breast cancer is absent. This is a study of limited interest.

Response: We acknowledge the reviewers viewpoint. Here we describe that we have analyzed several molecular markers and omics profiles. Since this manuscript is a cohort profile, we do not present the results of these analyses here, as this manuscript is meant to describe the data material gathered in this cohort. In the core of 100 samples we have additional data, as listed in the manuscript. We have both epidemiological and life-style data, as well as histopathological data. Furthermore, we have stained for several molecular markers and we have omics profiles both from blood samples and tumor tissue. We believe this subset is approaching the full systems epidemiological potential due to the wide range of available data. 

 

Reviewer 3

“The manuscript “Cohort profile: The Clinical and Multi-omic (CAMO) cohort, part of the Norwegian Women and Cancer (NOWAC) study” is basically a description of study design, data collection, and representativeness of the breast cancer cases of CAMO compared to the larger NOWAC study. The data collection is surely very unique.”

My suggestions to improve readability and clarity of the manuscript, and to focus on what is really the aim of this manuscript are the following (in order of the manuscript):

“In the manuscript the CAMO cohort includes women with breast cancer, please clarify if this is only invasive or also DCIS.”

Response: We highly appreciate that the reviewer finds the collected data to be very unique. Only women with invasive breast carcinomas are included in the CAMO cohort. To clarify this important issue, we have included the word «invasive» in the Abstract, p. 2, line 28, which now reads «..These study participants were diagnosed with invasive breast cancer…» and in the Introduction, p. 4 line 77-78, which now reads «…388 female patients with invasive breast cancer…».

“Were all women included before or after a first primary breast cancer?”

Response: All women in the CAMO-cohort were recruited to the cohort after primary breast cancer. Women in the NOWAC study were recruited regardless of cancer status at the time of recruitment. 

“How was dealt with women that had another cancer before their breast cancer or were there none?”

Response: From the cancer registry we have received the first cancer diagnosis regardless of cancer type. We can thus see which type of cancer was the participants' first type of cancer, and thus distinguish those participants from the rest. 

“The figures are quite informative. However, Figure 2 was not immediately clear to me. It would help to add the information related to A-E directly in text boxes with the figure.

 Is each dot 5 women? Then I think I only count 380 women?”

Response: Because of the amount of information related to A-E, we are not able to add the information directly into text boxes within the figure. Due to rounding by the software used to create the figures the amount of the dots is not exact. The lack of one and a half dot is due to rounding of the number of cases per year, resulting in the loss of a dot. 

“It is also not visible in the figure which proportion of women had second follow-up is it possible to somehow visualize this? I would otherwise refer to table 1 (see next comment).”

Response: Unfortunately, we are not able to clearly visualize the proportion of women who had a second follow-up in Figure 2. However, this information is presented in Table 1 following immediately after Fig. 2. In the manuscript. 

“In table 1 add the median (range) time between baseline and first and second follow-up. This information is especially needed given the claims in the discussion about the value of having a cohort with different times between cancer-free and after cancer diagnosis.”

Response: We have provided the requested information in line 247-248 of the Results section.

“Page 6 line 129/130. It is a pity that less than half of the women provided a blood sample before breast cancer diagnosis, more incident breast cancers would have been valuable. How does this proportion of incident and prevalent breast cancer (in relation to the blood sampling) in CAMO relates to NOWAC?”

We highly appreciate the reviewer’s comment. Please note that we have corrected the information provided in lines 130-133 to correspond with the information in Table 1: 53.2 % of the women provided a blood sample before diagnosis. Correspondingly, in the NOWAC Post-genome cohort (n=50 000), there are 3035 BC cases, of which 1888 (62%) gave a blood sample before the BC diagnosis. We agree with the reviewer that it would be very interesting and valuable to have more pre-diagnostic blood samples. However, as described in Materials and methods, section on Study population, the NOWAC study was a prospective cohort study started in 1991 which was expanded to include blood samples in 2003-2006. The samples were collected irrespective of any previous breast cancer diagnosis, and information on diagnosis of breast cancer were identified in retrospect through linkage to the Cancer Registry of Norway. This would result in a proportion of the patients having blood samples drawn after their diagnosis.

Page 7 line 145 “self-reporting of family history of breast cancer (mother, sister),”

 Please clarify. Was indeed only family history of mothers and sister reported, not of all first-degree relatives?

Response: That’s correct, only the family history of mother and sister where included. 

“Page 8 “To ensure diagnostic quality, we reanalyzed ER and PR for all breast cancers diagnosed before 2001”

 Which proportion of the data does this concern? There will also still be substantial variation between hospitals/ laboratories, so why did the authors not re-analyze receptors for all samples? For those that were not stained again, were these re-scored by one pathologist?”

Response: We appreciate the reviewer’s comment. As illustrated in Fig. 2, around 104 of all the breast cancers included in the cohort were diagnosed before 2001. The analyses of ER and PR were done as part of routine diagnostic evaluation at two closely collaborating pathology labs. However, all histopathological parameters included in the study have been re-evalutated by a breast pathologist (co-author L.M.). Since many of the older slides were of suboptimal quality and ER and PR were reported using scores no longer included in national guidelines (e.g. Allred score, different cut-offs) we found that all slides before 2001 should be restained and -analyzed, but all ER- and PR-stained slides were evaluated by the breast pathologist in the study. We have now included a line about this in the description of the clinical database on p. 8, line 162-163: «All histopathological data, including receptor status, were re-evaluated by a breast pathologist in the study.» 

Page 14 discussion: “Molecular markers in tumors have been analyzed for the entire cohort, whereas a core of approximately 100 samples is approaching the full systems epidemiology potential.”

Where can this number be found back in the text or tables? Figure 3 and tables 1-3 suggest that all data is available for almost all breast cancer cases. Does this only refer to the part on which there is also omics data? I strongly suggest including a table with omics information in the results section (not necessarily results of analyses, but at least showing which data is available for how many samples). It would also be informative to show how data availability is distributed over core breast cancer subgroups e.g. pre- versus post-menopausal and subtypes (at least ER positive versus negative).

We appreciate the reviewer’s comment. Please note as illustrated in Fig. 3 that information from national registries, questionnaires and medical records and collected tumor tissue and blood samples are available for all breast cancer cases included in the CAMO cohort. As stated in Material and methods, section on Analyses of blood samples, a series of omics analyses have already been carried out on a subset of these samples. This illustrates that the collected material available for the entire CAMO-cohort is suitable for omics analyses. However, to clarify the number of samples for which omics data already is available, we have included this information in line 191 and given a short summary in lines 221-223. Further, the breast cancer cases included in the pilot study have been described in great detail, including information on breast cancer subgroups etc., in the published article referred to in the present manuscript (ref. 16). We have rephrased lines 309-310 in the Discussion to clarify that information on these 100 samples can be found in ref. 16. In this cohort profile manuscript, our emphasis is on the CAMO cohort as a whole and its’ potential.

“The discussion is overall quite long, would consider reducing the parts on own previous results and on multiple dimensions. Related to previous results, a burning question would be how the CAMO dataset can help with the next steps, this is not really addressed.

The data collection is very unique, but still the sample set is small for multilevel analyses. Perhaps the discussion should be tuned down a bit on the promises for the future.”

Response: Since this is a Cohort-profile paper, and the data material in the CAMO cohort has already been used in other research project, we wanted to describe the previous research thoroughly in the discussion section of the manuscript. In line 325 to 335 we discuss the possible future use of the data material in the CAMO-cohort. 

We appreciate the reviewer’s suggestion on toning down the discussion on future use of the CAMO-cohort. However, regarding the sample size, we believe further studies using the CAMO cohort can generate hypothesis that can be tested in larger cohorts.

One of the most important pieces of information, which is basically table 2 only appears late in the discussion (p17 row 350-356), suggest moving this earlier in the discussion.

Response: We also address the information in Table 2 in the lines 254-269 under Results. 

“In CAMO, serial sampling of biological specimens was not carried out. To account for this, we have approached the time issue using group level data, exploiting the randomness of the length of follow-up time due to the study design of the NOWAC Post genome cohort.”

Indeed, the lack of serial sampling may be mentioned as a shortcoming. However, the current manuscript does not provide any support or evidence in the shown tables or figures that this short coming is covered by the study design and sampling over a longer period. Frankly I would debate that this is possible unless the study would be very large and the distribution of patient and tumor characteristics well balanced over time between samples and diagnosis of cancer.

Response: We appreciate the comment from the reviewer and recognize this shortcoming in our data. However, we have previously demonstrated that the samples can be used to investigate time trends. This was shown in ref. 18 and 20, which are part of our discussion section highlighting some of the potential uses of the data. In the provided references, assumptions and shortcomings of this design are also discussed.

“The NOWAC study was previously externally validated (28).” Please explain better, not clear what is meant here.

Response: It is important to examine the external validity i.e. the possibility to make inferences to the general population outside the study sample. In the study that we refer to the authors investigated three different methodological aspects of external validity regarding NOWAC; linkage to national registries, inquiry to non-responders and comparison between observed and expected cancer incidence rates. The authors conclude that the analysis revealed no major source of selection bias and that NOWAC cohort is representative of women living in Norway. 

We have included the phrase “… by Lund et al. 2003, revealing no major source of selection bias” on p. 17 line 367-368 to make this clearer. 

Appendix. Correct typo page 1 “as as”

Response: We have corrected the typo on page 1 in the Appendix.

The appendix explains the scoring of the TMAs but not of the original and newly stained whole slides for ER, PR, HER2.

Response: The scoring of ER, PR and HER2 is explained in the section on the clinical database, p. 8.

This sentence in the Appendix “Each researcher gives three scores to each core, reflecting staining density and/or intensity in stroma, tumor cytoplasm and tumor nucleus.” needs some rewriting. I guess not all markers stain in all areas (stroma, cytoplasm, nucleas) and is always either density (is this % of cells?) or intensity (is this vague to strong coloring?) scored?

Response: We appreciate the reviewer’s comment and have rewritten this paragraph to make it clearer, «Typically, each researcher scores the staining density, i.e. number or percentage of positive cells, and/or intensity…».

“A better explanation of how the data can be requested, even if restrictions apply, should be included in the data availability statement”.

Response: We have included the following paragraph in the data availability statement: “Due to the sensitivity of the data that has been collected in the CAMO cohort, both from the data material collected in NOWAC and the medical records as well as from the linkage to the Cancer Registry, the data cannot be placed in a public repository. Data cannot be shared publicly since the data is of a detailed and sensitive nature. However, data will be available upon request to NOWAC at nowac@uit.no”

 

Reviewer 4

“The authors present a cohort study, CAMO, which sounds excellent. CAMO is a substudy of a larger national cohort of individuals; these individuals developed breast cancer and were tracked over a period of time. There was a baseline questionnaire and for most patients 2 followup questionaires regarding various quality of life metrics. There were blood samples taken, sometime before and sometimes after the patients were diagnosed with breast cancer. The blood samples have been used for various molecular studies including mRNA and miRNA profiling and metabolomics. This sounds a great resource. Methods are described for all the profiling studies that have taken place. The paper describes the cohort (demographics, clinical/pathological features), and compared to the larger cohort, with few notable differences or findings. 

I was looking forward to a description of the longitudinal quality of life data or the molecular analysis of blood samples or tissue samples, or the attempt at ‘systems epidemiological analysis’ but this was described as published data in the discussion or indicated as the potential for such a resource. I understand the value of describing a cohort and its potential for answering research questions before research has been done, but this seems the wrong way around. So I am puzzled as to why this is being submitted for publication at this point. Might be better to incorporate some data of interest to the readership (QoL data or tumour profiling studies that are in progress). Might be better submitted to a Biorepositories focused journal.”

Response: We appreciate the reviewer’s comment. Cohort profiles are articles that describe ongoing research cohorts, and, in brief, cohort profiles will describe large, collaborative prospective studies that identify a group of participants and follow them for long periods. 

In contrast to research papers, that are traditional results papers and should address a specific research question, the reason to publish cohort profile papers is to provide information on a cohort’s establishment. The information that is published in cohort profile papers goes beyond what can reasonably be described in the methods section of a research paper. Another reason to publish a cohort profile paper is to advise other researchers of existing datasets and opportunities for collaboration.

We think that the advantage to publish our Cohort profile paper to PLOS ONE is so that anyone interests can easily access it when appraising studies that arise from it. The data material in the CAMO cohort has already been used in other research project, hence why we describe the research in the discussion section of the manuscript.

---

## [Decision Letter · Decision Letter 1]

18 Jan 2023

Cohort profile: The Clinical and Multi-omic (CAMO) cohort, part of the Norwegian Women and Cancer (NOWAC) study

PONE-D-22-25688R1

Dear Dr. Delgado,

We’re pleased to inform you that your manuscript has been judged scientifically suitable for publication and will be formally accepted for publication once it meets all outstanding technical requirements.

Kind regards,

Alvaro Galli

Academic Editor

PLOS ONE

Additional Editor Comments (optional):

Reviewers' comments:

Reviewer's Responses to Questions

**Comments to the Author**

1. If the authors have adequately addressed your comments raised in a previous round of review and you feel that this manuscript is now acceptable for publication, you may indicate that here to bypass the “Comments to the Author” section, enter your conflict of interest statement in the “Confidential to Editor” section, and submit your "Accept" recommendation.

Reviewer #3: All comments have been addressed

2. Is the manuscript technically sound, and do the data support the conclusions?

Reviewer #3: Yes

3. Has the statistical analysis been performed appropriately and rigorously? 

Reviewer #3: Yes

4. Have the authors made all data underlying the findings in their manuscript fully available?

Reviewer #3: No

5. Is the manuscript presented in an intelligible fashion and written in standard English?

Reviewer #3: Yes

6. Review Comments to the Author

Reviewer #3: The authors have sufficiently addressed the feedback. Personally I would have reshaped the discussion a bit more and made some further attempts to clarity the data (e.g. being clearer on how many women had a previous cancer or not, and adding a footnote to figure 2 explaining that the number of dots is less due to rounding), but the difference in viewpoint of what should and should not be included in this manuscript does in my view not hamper publication of the manuscript.

7. PLOS authors have the option to publish the peer review history of their article (what does this mean?). If published, this will include your full peer review and any attached files.

Reviewer #3: No

---

## [Editor Report · Acceptance letter]

26 Jan 2023

PONE-D-22-25688R1 

Cohort profile: The Clinical and Multi-omic (CAMO) cohort, part of the Norwegian Women and Cancer (NOWAC) study 

Dear Dr. Delgado:

I'm pleased to inform you that your manuscript has been deemed suitable for publication in PLOS ONE. Congratulations! Your manuscript is now with our production department. 

Kind regards, 

on behalf of

Dr. Alvaro Galli 

Academic Editor

PLOS ONE